

# How to GAN LHC events

**Anja Butter, Tilman Plehn and Ramon Winterhalder⋆**

Institut für Theoretische Physik, Universität Heidelberg, Germany

⋆ winterhalder@thphys.uni-heidelberg.de

## Abstract

Event generation for the LHC can be supplemented by generative adversarial networks, which generate physical events and avoid highly inefficient event unweighting. For top pair production we show how such a network describes intermediate on-shell particles, phase space boundaries, and tails of distributions. In particular, we introduce the maximum mean discrepancy to resolve sharp local features. It can be extended in a straightforward manner to include for instance off-shell contributions, higher orders, or approximate detector effects.

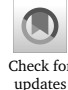
**Content**

## 1  Introduction

First-principle simulations are a key ingredient to the ongoing success of the LHC, and they are crucial for further developing it into a precision experiment testing the structure of the Standard Model and its quantum field theory underpinnings. Such simulations of the hard scattering process, QCD activity, hadronization, and detector effects are universally based on

Monte Carlo methods. These methods come with structural challenges, for example related to an efficient coverage of the high-dimensional phase space, event unweighting, or complex and hence slow detector simulations. Some of these problems might be alleviated when we add a new direction, like machine learning techniques, to our tool box. While we should not expect them to magically solve all problems, we have seen that modern machine learning can trigger significant progress in LHC physics. The reason for our optimism related to event generation are generative adversarial networks or GANs [1], which have shown impressive performance in tasks like the generation of images, videos or music.

From the experimental side the detector simulation is the most time-consuming aspect of LHC simulations, and promising attempts exist for describing the behavior of the calorimeter with the help of generative networks [2–7]. On the theory side, we know that the parton shower can be described by a neural network [8–11]. It has been shown that neural networks can help with phase space integration [12,13] and with LHC event simulations [14–16]. One open question is why the GAN setup of Ref. [14] does not properly work and is replaced by a variational autoencoder with a density information buffer. Another challenge is how to replace the ad-hoc $Z$-constraint in the loss function of Ref. [15] by a generalizable approach to on-shell resonances. This problem of intermediate resonances is altogether avoided in Ref. [16]. It remains to be shown how GANs can actually describe realistic multi-particle matrix elements over a high-dimensional phase space in a flexible and generalizable manner.

In this paper we show how we can efficiently GAN* the simulation of the $2 \rightarrow 6$ particle production process

$$pp \rightarrow t\bar{t} \rightarrow (bq\bar{q}')\,(\bar{b}\bar{q}q'), \tag{1}$$

describing all intermediate on-shell states with Breit-Wigner propagators and typical width-to-mass ratios of few per-cent. We will focus on a reliable coverage of the full phase space, from simple momentum distributions to resonance peaks, strongly suppressed tails, and phase space boundaries.

Given this new piece of the event simulation puzzle through fast neural networks it should in principle be possible to add parton showers, possibly including hadronization, and detector effects to a full machine learning description of LHC events. Including higher-order corrections is obviously possible and should lead to ever higher gains in computing time, assuming higher-orders are included in the training data. The interesting question then becomes where established methods might benefit from the fast and efficient machine learning input. Alternatively, we can replace the Monte Carlo event input and instead generate reconstructed LHC events and use them to enhance analyses or to study features of the hard process. Obviously, the GAN approach also allows us to combine information from actual data with first-principles simulations in a completely flexible manner.

Our paper consists of two parts. In Sec. 2 we start by reviewing some of the features of phase space sampling with Monte Carlo methods and introducing GANs serving the same purpose. We then add the MMD and describe how its been used to describe intermediate resonances. In Sec. 3 we apply the combined GAN-MMD network to top pair production with subsequent decays and show that it describes the full phase space behavior, including intermediate on-shell particles.

---

*From 'to GAN', in close analogy to the verbs taylor, google, and sommerfeld.

## 2 Phase space generation

As a benchmark model throughout this paper we rely on top pair production with an intermediate decay of two $W$-bosons

$$pp \to t\bar{t} \to (bW^-)(\bar{b}W^+) \to (bf_1\bar{f}'_1)(\bar{b}f_2\bar{f}'_2)\,, \tag{2}$$

illustrated in Fig. 1. If we assume that the masses of all final-state particles are known, as this can be extracted from the measurement, this leaves us with 18 degrees of freedom, which energy-momentum conservation reduces to a 14-dimensional phase space. In addition, our LHC simulation has to account for the 2-dimensional integration over the parton momentum fractions.

In this section we will briefly review how standard methods describe such a phase space, including the sharp features of the intermediate on-shell top quarks and $W$-boson. The relevant area in phase space is determined by the small physical particle widths and extends through a linearly dropping Breit-Wigner distribution, where it eventually needs to include off-shell effects. We will then show how a generative adversarial network can be constructed such that it can efficiently handle these features as well.

### 2.1 Standard Monte Carlos

For the hard partonic process we denote the incoming parton momenta as $p_{a,b}$ and the outgoing fermion momenta as $p_f$. The partonic cross section and the 14-dimensional phase-space integration for six external particles can be parametrized as

$$\int d\sigma = \int d\Phi_{2\to6}\, \frac{|\mathcal{M}(p_a,p_b;p_1,\dots,p_6)|^2}{2\hat{s}}\,,$$

$$\text{with} \quad d\Phi_{2\to6} = (2\pi)^4 \delta^{(4)}(p_a + p_b - p_1 - \cdots - p_6) \prod_{f=1}^{6} \frac{d^3 p_f}{(2\pi)^3} \frac{1}{2p_f^0} \Bigg|_{p_f^0 = \sqrt{\vec{p}_f^2 + m_f^2}}\,. \tag{3}$$

To cope with the high dimensionality of the integral we adopt advanced Monte Carlo techniques. The integral of a function $f(x)$ over a volume $V$ in $\mathbb{R}^d$

$$I = \int_V d^d x\, f(x) \tag{4}$$

can be approximated with the help of $N$ random numbers $x_i$ distributed according to a normalized density function $\rho(x)$

$$\int_V d^d x\, \rho(x) = 1\,, \tag{5}$$

Figure 1: Sample Feynman diagram contributing to top pair production, with intermediate on-shell particles labelled.

such that

$$I \approx S_N = \frac{1}{N} \sum_{i=1}^{N} \frac{f(x_i)}{\rho(x_i)} \ . \tag{6}$$

For sufficiently large $N$ the variance of this integral scales like

$$\sigma^2 \approx \frac{1}{N-1} \left( \frac{1}{N} \sum_{i=1}^{N} \frac{f(x_i)^2}{\rho(x_i)^2} - S_N^2 \right) , \tag{7}$$

which means that it can be minimized by an appropriate choice of $\rho(x)$. This requires $\rho(x)$ to be large in regions where the integrand is large, for instance

$$\rho(x) = \frac{|f(x)|}{\int_V \mathrm{d}^d x \, f(x)} \ . \tag{8}$$

This method of choosing an adequate density is called importance sampling. There are several implementations available, one of the most frequently used is Vegas [17, 18].

A major challenge in particle physics applications is that multi-particle amplitudes in the presence of kinematic cuts typically have dramatic features. Our phase space sampling not only has to identify the regions of phase space with the leading contribution to the integral, but also map its features with high precision. For instance, the process illustrated in Fig. 1 includes narrow intermediate on-shell particles. Around a mass peak with $\Gamma \ll m$ they lead to a sharp Breit-Wigner shape of the transition amplitude. A standard way of improving the integration is to identify the invariant mass variable $s$ where the resonance occurs and switch variables to

$$\int \mathrm{d}s \frac{F(s)}{(s-m^2)^2 + m^2\Gamma^2} = \frac{1}{m\Gamma} \int \mathrm{d}z \, F(s) \quad \text{with} \quad z = \arctan \frac{s-m^2}{m\Gamma} \ . \tag{9}$$

This example illustrates how phase space mappings, given some knowledge of the structure of the integrand, allow us to evaluate high-multiplicity scattering processes.

Finally, in LHC applications we are typically not interested in an integral like the one shown in Eq.(3). Instead, we want to simulate phase space configurations or events with a probability distribution corresponding to a given hard process, shower configuration, or detector smearing. This means we have to transfer the information included in the weights at a given phase space point to a phase space density of events with uniform weight. The corresponding unweighting procedure computes the ratio of a given event weight to the maximum event weights, probes this ratio with a random number, and in turn decides if a phase space point or event remains in the sample, now with weight one. This procedure is highly inefficient.

Summarizing, the challenge for a machine learning approach to phase space sampling is: mimic importance sampling, guarantee a precise mapping of narrow patterns, and avoid the limited unweighting efficiency.

## 2.2 Generative adversarial network

The defining structural elements of generative adversarial networks or GANs are two competing neural networks, where the generator network $G$ tries to mimic the data while the discriminator network $D$ is trained to distinguish between generated and real data. The two networks play against each other, dynamically improving the generator by searching for parameter regions where the generator fails and adjusting its parameters there.

To start with, both networks are initialized with random values so that the generator network induces a underlying random distribution $P_G(x)$ of an event or phase space configuration

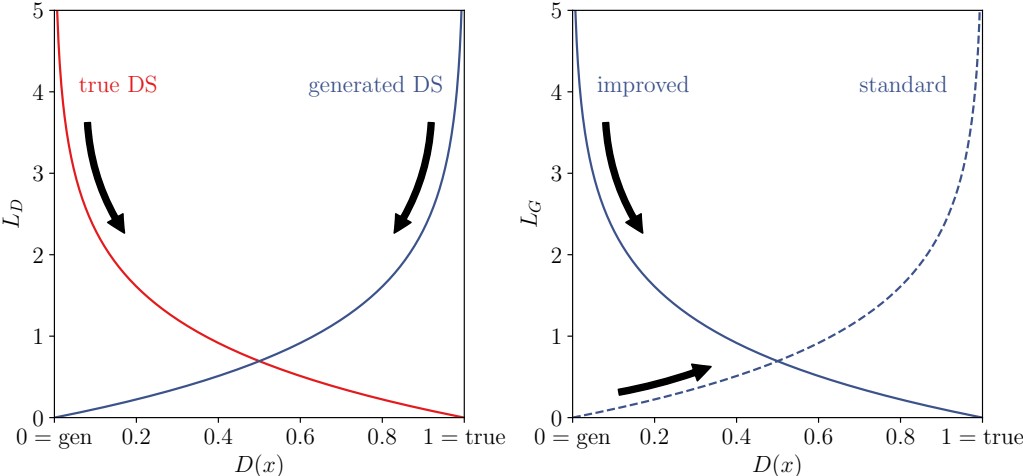

Figure 2: Discriminator and generator losses as a function of the value assigned by the discriminator. The red line indicates batches from the true distribution, the blue lines batches from a generated distribution. The arrows indicate the direction of the training.

$x$, typically organized with the same dimensionality as the (phase) space we want to generate. Now the discriminator network compares two distributions, the true distribution $P_T(x)$ and the generated distribution $P_G(x)$. From each of the two distributions we provide batches of phase space configurations $\{x_T\}$ and $\{x_G\}$ sampled from $P_T$ or $P_G$, respectively. Here the sets $\{x_{T,G}\}$ are batches of events sampled from the training or generated data.

The discriminator output $D(x) \in (0,1)$ is trained to give $D = 1$ for each point in a true batch and $D = 0$ for the each point in the generated and hence not true batch. We can enhance the sensitivity for $D \to 0$ by evaluating the variable $-\log D(x) \in (\infty, 0)$ instead of $D(x)$ in the expectation value

$$\left\langle -\log D(x) \right\rangle_x = -\frac{1}{N_x} \sum_{x \in \text{batch}} \log D(x), \tag{10}$$

where $N_x$ is the batch size. For a correctly labelled true sample this expectation value gives zero. The loss function is defined such that it becomes minimal when the discriminator correctly predicts the true and generated batches

$$L_D = \left\langle -\log D(x) \right\rangle_{x \sim P_T} + \left\langle -\log(1 - D(x)) \right\rangle_{x \sim P_G}. \tag{11}$$

The symbol $x \sim P$ indicates phase space configurations sampled from $P$. In the GAN application this discriminator network gets successively re-trained for a fixed truth $P_T(x)$ but evolving $P_G(x)$, as illustrated in the left panel of Fig. 2. We can compute the discriminator loss in the limit where the generator has produced a perfect image of the true distribution. In this case the discriminator network will give $D = 0.5$ for each point $x$ and the result becomes $L_D = -2 \log 0.5 \approx 1.4$.

The generator network starts from random noise and transforms it into a distribution $P_G(x)$. For this it relies on the function $D(x)$, which encodes the truth information. Following Eq.(11) this means we can maximize its second term in the training of the generator network. It turns out that it is numerically more efficient to instead minimize the generator loss

$$L_G = \left\langle -\log D(x) \right\rangle_{x \sim P_G}. \tag{12}$$

In the right panel of Fig. 2 we see how this assignment leads to larger gradients away from the true configurations.

The key to the GAN training is the alternating training of the generator and discriminator networks with their respective loss functions given in Eq.(11) and Eq.(12). Here, the balance between generator and discriminator is crucial. On the one hand, the generator can only be as good as the discriminator which defines the level of similarity between true and generated data. On the other hand, a perfect discriminator leads to a vanishing loss function, which reduces the gradient and slows down the training. This interplay of the two networks often leads to stability issues in the training [19]. A common way to stabilize networks are noise-induced regularization methods, or equivalently including a penalty on the gradient for the discriminator variable $D(x)$ [20]. Specifically, we apply the gradient to the monotonous logit function

$$\phi(x) = \log \frac{D(x)}{1-D(x)} \qquad \Rightarrow \qquad \frac{\partial \phi}{\partial x} = \frac{1}{D(x)} \frac{1}{1-D(x)} \frac{\partial D}{\partial x}, \tag{13}$$

enhancing its sensitivity in the regions $D \to 0$ or $D \to 1$. The penalty applies to regions where the discriminator loss leads to a wrong prediction, $D \approx 0$ for a true batch or $D \approx 1$ away from the truth. This means we add a term to the discriminator loss and obtain the regularized Jensen-Shannon GAN objective [20]:

$$L_D \to L_D + \lambda_D \big\langle (1-D(x))^2 \, |\nabla \phi|^2 \big\rangle_{x \sim P_T} + \lambda_D \big\langle D(x)^2 \, |\nabla \phi|^2 \big\rangle_{x \sim P_G}, \tag{14}$$

with a properly chosen variable $\lambda_D$. The pre-factors $(1-D)^2$ and $D^2$ indeed ensure that for a properly trained discriminator this additional contribution vanishes. Another method to avoid instabilities in the training of the GAN is to use the Wasserstein distance [21,22] but our tests have shown that Eq.(14) works better in our case.

As a side remark, another common type of neural network used for generative problems are variational autoencoders (VAE). They perform a dimensional reduction of the input data — often an image — to create a latent representation. The autoencoder is trained to minimize the difference between input and inferred image, where a variational autoencoder requires the components of the latent representation to follow a Gaussian. If we then insert Gaussian random numbers for the latent representation, the decoder generates new images with the same characteristics as the training data. While VAEs can be used to generate new data samples, a key component is the latent modelling and the marginalization of unnecessary variables, which is not a problem in generating LHC events.

## 2.3 Loss functions for intermediate particles

A particular challenge for our phase space GAN will be the reconstruction of the $W$ and top masses from the final-state momenta. For instance, for the top mass the discriminator and generator have to probe a 9-dimensional part of the phase space, where each direction covers several 100 GeV to reproduce a top mass peak with a width of $\Gamma_t = 1.5$ GeV. Following the discussion of the Monte Carlo methods in Sec. 2.1 the question is how we can build an analogue to the phase space mappings for Monte Carlos. Assuming that we know which external momenta can form a resonance we explicitly construct the corresponding invariant masses and give them to the neural network to streamline the comparison between true and generated data. We emphasize that this is significantly less information than we use in Eq.(9), because the network still has to learn the intermediate particle mass, width, and shape of the resonance curve.

A suitable tool to focus on a low-dimensional part of the full phase space is the maximum mean discrepancy (MMD) [23]. The MMD is a kernel-based method to compare two samples

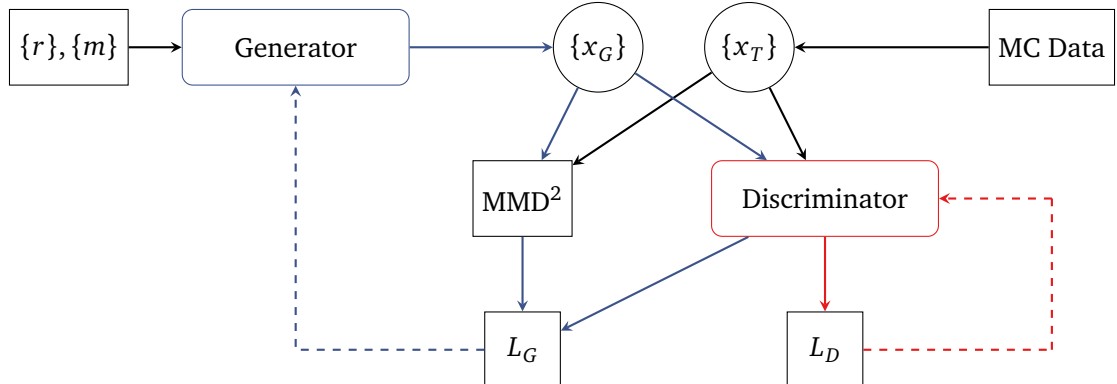

Figure 3: Schematic diagram for our GAN. The input $\{r\}$ and $\{m\}$ describe a batch of random numbers and the masses of the external particles, and $\{x\}$ denotes a batch of phase space points sampled either from the generator or the true data. The blue (red) and arrows indicate which connections are used in the training of the generator (discriminator).

drawn from different distributions. Using one batch of true data points and one batch of generated data points, it computes a distance between the distributions as

$$\text{MMD}^2(P_T, P_G) = \left\langle k(x,x')\right\rangle_{x,x'\sim P_T} + \left\langle k(y,y')\right\rangle_{y,y'\sim P_G} - 2\left\langle k(x,y)\right\rangle_{x\sim P_T, y\sim P_G}, \tag{15}$$

where $k(x,y)$ can be any positive definite kernel function. Obviously, two identical distributions lead to $\text{MMD}(P,P) = 0$ in the limit of high statistics. Inversely, if $\text{MMD}(P_T, P_G) = 0$ for randomly sampled batches the two distributions have to be identical $P_T(x) = P_G(x)$. The shape of the kernels determines how local the comparison between the two distributions is evaluated. Two examples are Gaussian or Breit-Wigner kernels

$$k_{\text{Gauss}}(x,y) = \exp -\frac{(x-y)^2}{2\sigma^2} \qquad \text{or} \qquad k_{\text{BW}}(x,y) = \frac{\sigma^2}{(x-y)^2 + \sigma^2}, \tag{16}$$

where the hyperparameter $\sigma$ determines the resolution. For an optimal performance it should be of the same order of magnitude as the width of the feature we are trying to learn. If the resonance and the kernel width become too narrow, we can improve convergence by including several kernels with increasing widths to the loss function. The shape of the kernel has nothing to do with the shape of the distributions we are comparing. Instead, the choice between the exponentially suppressed Gaussian and the quadratically suppressed Breit-Wigner determines how well the MMD accounts for the tails around the main feature. As a machine learning version of phase space mapping we add this MMD to the generator loss

$$L_G \rightarrow L_G + \lambda_G \, \text{MMD}^2, \tag{17}$$

with another properly chosen variable $\lambda_G$.

Similar efforts in using the MMD to generate events have already been done in [24–26] and has also been extended to a adversarial MMD version or MMD-GAN [27–29], in which the MMD kernel is learned by another network.

In Fig. 3 we show the whole setup of our network. It works on batches of simulated parton-level events, or unweighted event configurations $\{x\}$. The input for the generator are batches of random numbers $\{r\}$ and the masses $\{m\}$ of the final state particles. Because of the random input a properly trained GAN will generate statistically independent events reflecting the learned patterns of the training data. For both the generator and the discriminator we use a

Table 1: Details for our GAN setup.

| Parameter | Value |
|---|---|
| Input dimension G | $18 + 6$ |
| Layers | 10 |
| Units per layer | 512 |
| Trainable weights G | 2382866 |
| Trainable weights D | 2377217 |
| $\lambda_D$ | $10^{-3}$ |
| $\lambda_G$ | 1 |
| Batch size | 1024 |
| Epochs | 1000 |
| Iterations per epoch | 1000 |
| Training time | 26h |
| Size of trainings data | $10^6$ |

10-layer MLP with 512 units each, the remaining network parameters are given in Tab. 1. The main structural feature of the competing networks is that the output of the discriminator, $D$, is computed from the combination of true and generated events and is needed by the generator network. The generator network combines the information from the discriminator and the MMD in its loss function, Eq.(17). The learning is done when the distribution of generated unweighted events $\{x_G\}$ and true Monte-Carlo events $\{x_T\}$ are essentially identical. We again emphasize that this construction does not involve weighted events.

## 3  Machine-learning top pairs

A sample Feynman diagram for our benchmark process

$$pp \rightarrow t\bar{t} \rightarrow (bq\bar{q}')\,(\bar{b}\bar{q}q'), \tag{18}$$

is shown in Fig. 1. For our analysis we generate 1 million samples of the full $2 \rightarrow 6$ events as training data sample with MG5aMCNLO [30]. The intermediate tops and $W$-bosons allow us to reduce the number of Feynman diagrams by neglecting proper off-shell contributions and only including the approximate Breit-Wigner propagators. Our results can be directly extended to a proper off-shell description [31–33], which only changes the details of the subtle balance in probing small but sharp on-shell contributions and wide but flat off-shell contributions. Similarly, we do not employ any detector simulation, because this would just wash out the intermediate resonances and diminish our achievement unnecessarily.

Because we do not explicitly exploit momentum conservation our final state momenta are described by 24 degrees of freedom. Assuming full momentum conservation would for instance make it harder to include approximate detector effects. These 24 degrees of freedom can be reduced to 18 when we require the final-state particles to be on-shell. While it might be possible for a network to learn the on-shell conditions for external particles, we have found that learning constants like external masses is problematic for the GAN setup. Instead, we use on-shell relations for all final-state momenta in the generator network.

Combining the GAN with the MMD loss function of Eq.(17) requires us to organize the generator input in terms of momenta of final-state particles. With the help of a second input to the generator, namely a 6-dimensional vector of constant final-state masses, we enhance the

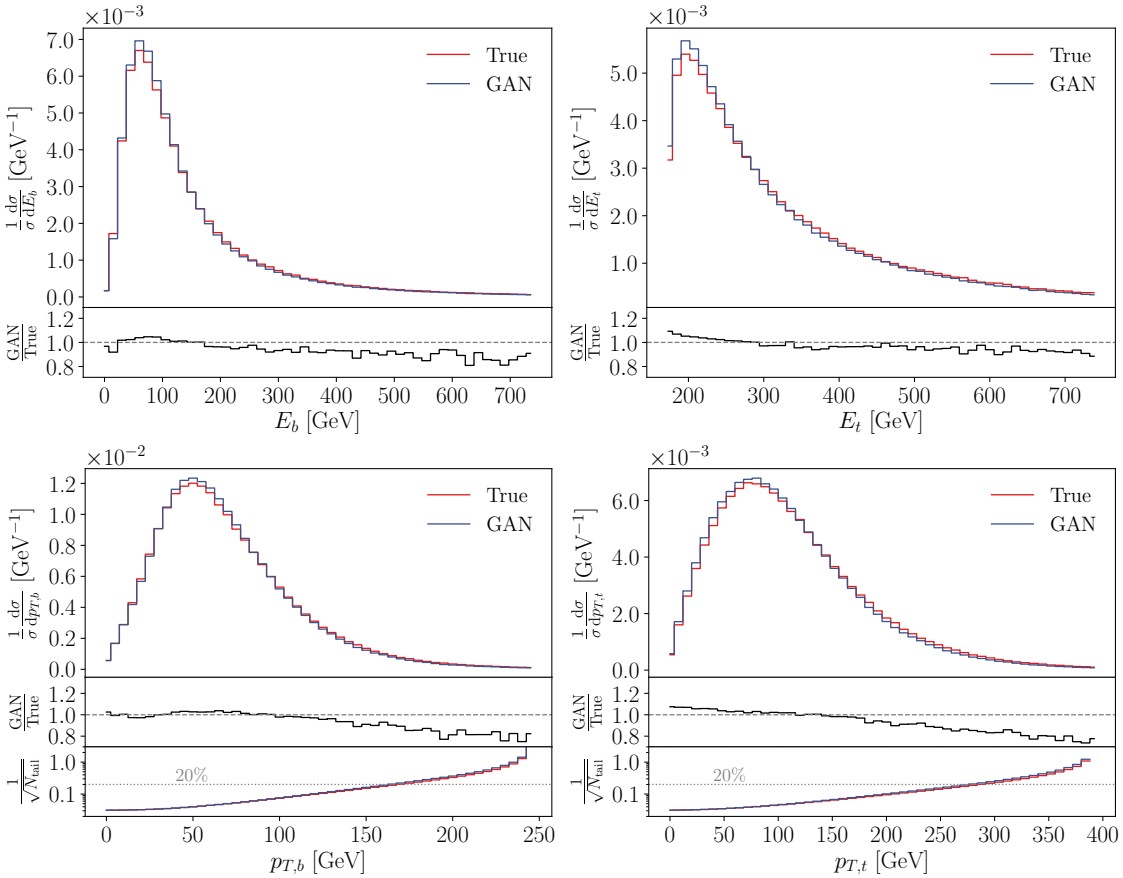

Figure 4: Energy (top) and transverse momentum (bottom) distributions of the final-state $b$-quark (left) and the decaying top quark (right) for MC truth (blue) and the GAN (red). The lower panels give the bin-wise ratio of MC truth to GAN distribution. For the $p_T$ distributions we show the relative statistic uncertainty on the cumulative number of events in the tail of the distribution for our training batch size.

18-dimensional input to six 4-vectors. This way we describe all final-state particles, denoted as $\{x_G\}$ in Fig. 3, through an array

$$x = \{p_1, p_2, p_3, p_4, p_5, p_6\}, \qquad (19)$$

where we fix the order of the particles within the events. This format corresponds to the generated unweighted truth events $\{x_T\}$ from standard LHC event simulators. In particular, we choose the momenta such that

$$p_{W^-} = p_1 + p_2, \qquad p_{W^+} = p_4 + p_5, \qquad p_{\bar{t}} = p_1 + p_2 + p_3, \qquad p_t = p_4 + p_5 + p_6. \qquad (20)$$

For the on-shell states we extract the resonances from the full phase space and use those to calculate the MMD between the true and the generated mass distributions. This additional loss is crucial to enhance the sensitivity in certain phase space regions allowing the GAN to learn even sharp feature structures.

**Flat distributions**

To begin with, relatively flat distributions like energies, transverse momenta, or angular correlations should not be hard to GAN [14–16]. As examples, we show transverse momentum

and energy distributions of the final-state $b$-quarks and the intermediate top quarks in Fig. 4. The GAN reproduces the true distributions nicely even for the top quark, where the generator needs to correlate the four-vectors of three final-state particles.

To better judge the quality of the generator output we show the ratio of the true and generated distributions in the lower panels of each plot, for instance $E_b^{(G)}/E_b^{(T)}$ where $E_b^{(G,T)}$ is computed from the generated and true events, respectively. The bin-wise difference of the two distributions increases to around 20% only in the high-$p_T$ range where the GAN suffers from low statistics in the training sample. To understand this effect we also quantify the impact of the training statistics per batch for the two $p_T$-distributions. In the set of third panels we show the relative statistical uncertainty on the number of events $N_{\text{tail}}(p_T)$ in the tail above the quoted $p_T$ value. The relative statistical uncertainty on this number of events is generally given by $1/\sqrt{N_{\text{tail}}}$. For the $p_{T,b}$-distribution the GAN starts deviating at the 10% level around 150 GeV. Above this value we expect around 25 events per batch, leading to a relative statistical uncertainty of 20%. The top kinematics is harder to reconstruct, leading to a stronger impact from low statistics. Indeed, we find that the generated distribution deviates by 10% around $p_{T,t} \gtrsim 250$ GeV where the relative statistic uncertainty reaches 15%.

We emphasize that this limitation through training statistics is expected and can be easily corrected for instance by slicing the parameter in $p_T$ and train the different phase space regions separately. Alternatively, we can train the GAN on events with a simple re-weighting, for example in $p_T$, but at the expense of requiring a final unweighting step.

**Phase space coverage**

To illustrate that the GAN populates the full phase space we can for instance look at the azimuthal coordinates of two final-state jets in Fig. 5. Indeed, the generated events follow the expected flat distribution and correctly match the true events.

Furthermore, we can use these otherwise not very interesting angular correlations to illustrate how the GAN interpolates and generates events beyond the statistics of the training data. In Fig. 6 we show the 2-dimensional correlation between the azimuthal jet angles $\phi_{j_1}$ and $\phi_{j_2}$. The upper-left panel includes 1 million training events, while the following three panels show an increasing number of GANed events, starting from 1 million events up to 50 million events. As expected, the GAN generates statistically independent events beyond the sample size of the training data and of course covers the entire phase space.

**Resonance poles**

From Ref. [12] we know that exactly mapping on-shell poles and tails of distributions is a challenge even for simple decay processes. Similar problems can be expected to arise for phase space boundaries, when they are not directly encoded as boundaries of the random number input to the generator. Specifically for our $t\bar{t}$ process, Ref. [14] finds that their GAN setup does not reproduce the phase space structure. The crucial task of this paper is to show how well our network reproduces the resonance structures of the intermediate narrow resonances. In Fig. 7 we show the effect of the additional MMD loss on learning the invariant mass distributions of the intermediate $W$ and top states. Without the MMD, the GAN barely learns the correct mass value, in complete agreement with the findings of Ref. [15]. Adding the MMD loss with default kernel widths of the Standard Model decay widths drastically improves the results, and the mass distribution almost perfectly matches the true distribution in the $W$-case. For the top mass and width the results are slightly worse, because its invariant mass needs to be reconstructed from three external particles and thus requires the generator to correlate more variables. This gets particularly tricky in our scenario, where the $W$-peak reconstruction directly affects the top peak. We can further improve the results by choosing a bigger batch

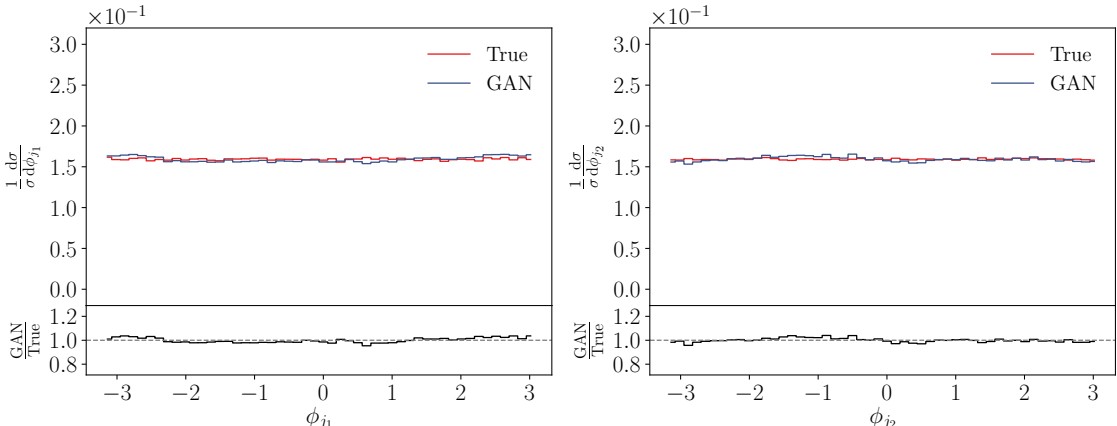

Figure 5: $\phi$ distributions of $j_1$ and $j_2$. The lower panels give the bin-wise ratio of MC truth to GAN distribution.

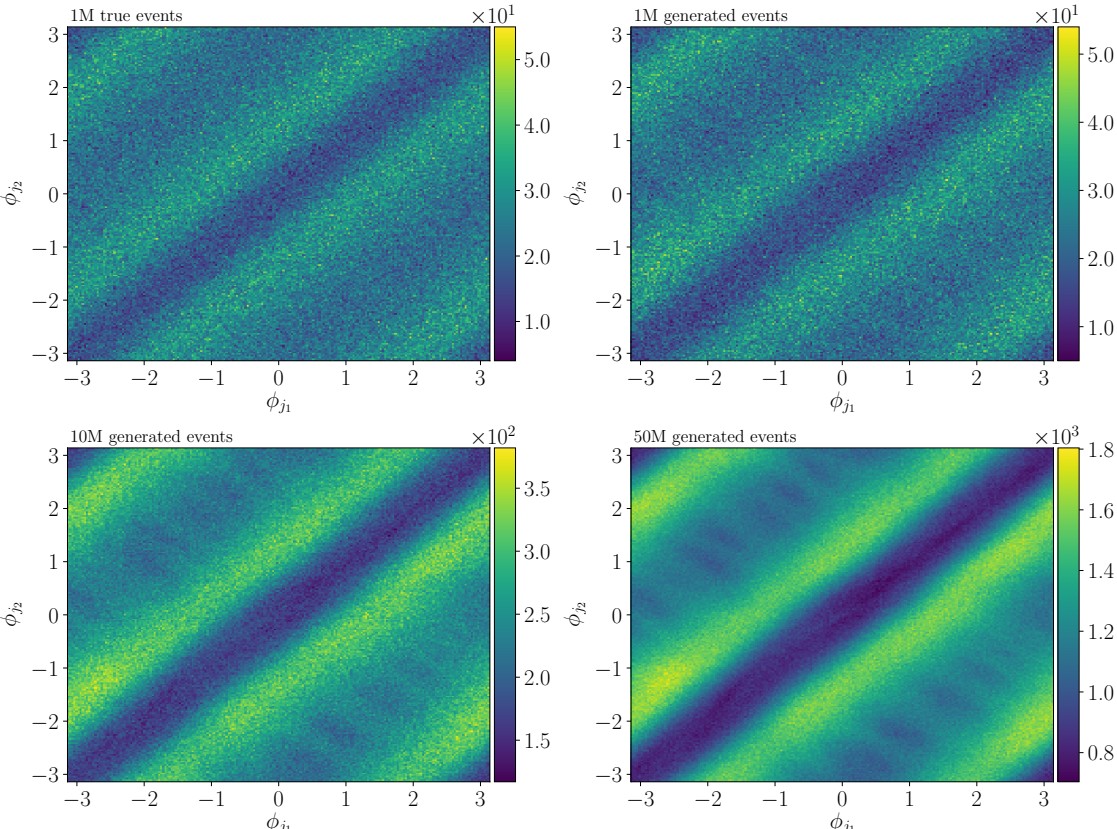

Figure 6: Correlation between $\phi_{j_1}$ and $\phi_{j_2}$ for 1 million true events (upper left) and for 1 million, 10 million, and 50 million GAN events.

size as this naturally enhances the power of the MMD loss. However, bigger batch sizes leads to longer training times and bigger memory consumption. In order to keep the training time on responsible level, we limited our batch size to 1024 events per batch. As already pointed out, the results are not perfect in this scenario, especially for the top invariant mass, however, we can clearly see the advantages of adding the MMD loss.

To check the sensitivity of the kernel width on the results, we vary it by factors of $\{1/4, 4\}$.

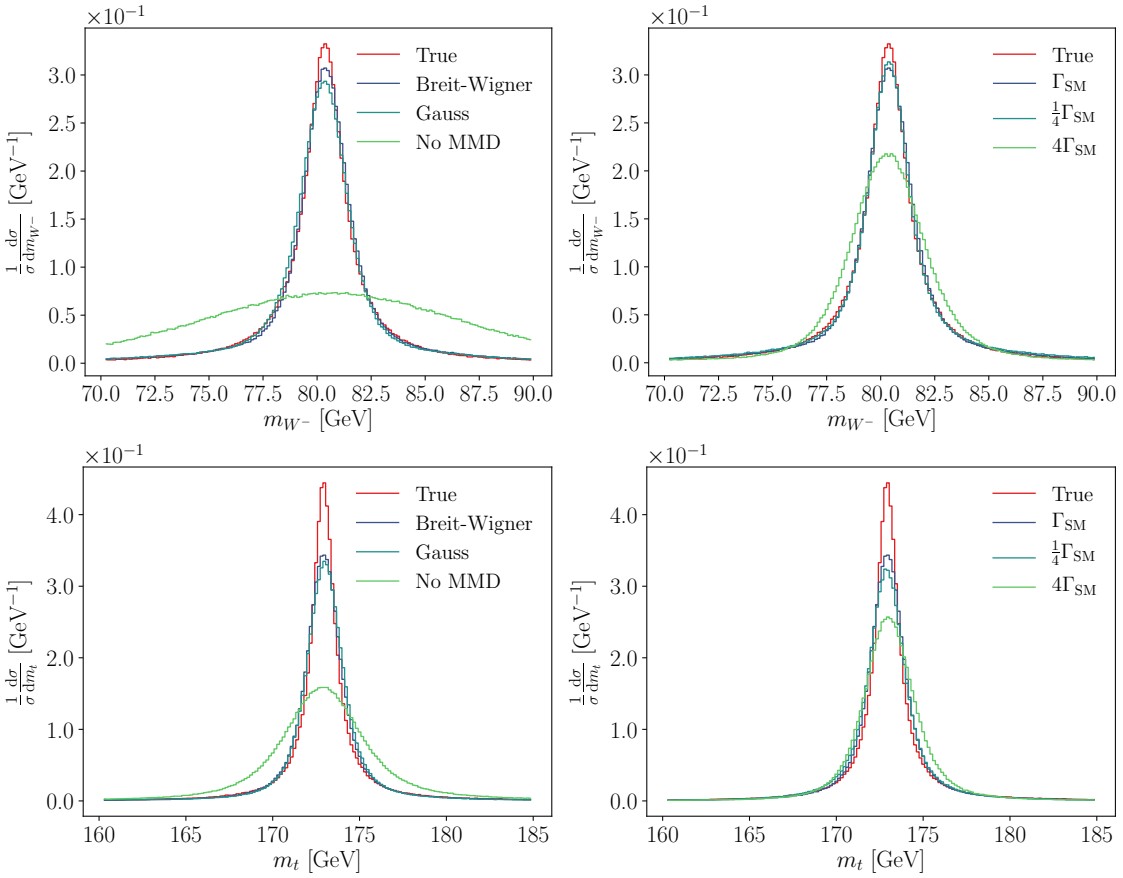

Figure 7: Comparison of different kernel functions (left) and varying widths (right) and their impact on the invariant mass of W boson (top) and top quark (bottom).

As can be seen in the lower panels of both distributions, increasing the resolution of the kernel or decreasing the kernel width hardly affects the network performance. On the other hand, increasing the width decreases the resolution and leads to too broad mass peaks. Similarly, if we switch from the default Breit-Wigner kernel to a Gaussian kernel with the same width we find identical results. This means that the only thing we need to ensure is that the kernel can resolve the widths of the analyzed features.

We emphasize again that we do not give the GAN the masses or even widths of the intermediate particles. This is different from Ref. [15], which tackles a similar problem for the $Z \to \ell\ell$ resonance structure and uses an explicit mass-related term in the loss function. We only specify the two final-state momenta for which the invariant mass can lead to a sharp phase space structure like a mass peak, define a kernel like those given in Eq.(16) with sufficient resolution and let the GAN do the rest. This approach is even more hands-off than typical phase space mappings employed by standard Monte Carlos.

**Correlations**

Now that we can individually GAN all relevant phase space structures in top pair production, it remains to be shown that the network also covers all correlations. A simple test is 4-momentum conservation, which is not guaranteed by the network setup. In Fig. 8, we show the sums of the transverse components of the final-state particle momenta divided by the sum of their absolute values. As we can see, momentum conservation at the GAN level is satisfied at the order of 2%.

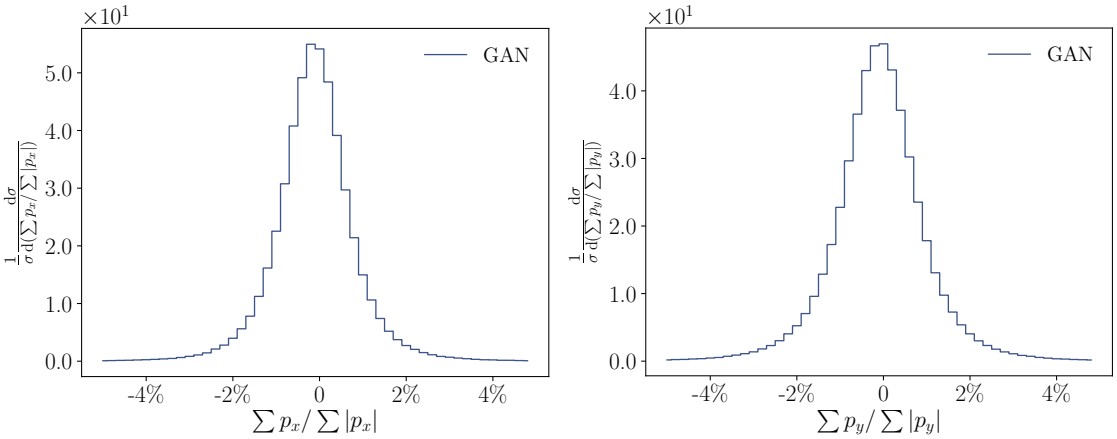

Figure 8: Sum of all $p_x$ ($p_y$) momenta divided by the sum of the absolute values in the left (right) panel, testing how well the GAN learns momentum conservation.

Finally, in Fig. 9 we show 2-dimensional correlations between the transverse momenta of the outgoing $b$-quark and the intermediate top for the true (left) and GAN events (right). The phase space structure encoded in these two observables is clearly visible, and the GAN reproduces the peak in the low-$p_T$ range, the plateau in the intermediate range, and the sharp boundary from momentum conservation in the high-$p_T$ range. To allow for a quantitative comparison of true and generated events we show the bin-wise asymmetry in the lower left panel. Except for the phase space boundary the agreement is essentially perfect. The asymmetry we observe along the edge is a result from very small statistics. For an arbitrarily chosen $p_T$ value of 100 GeV the deviations occur for $p_{T,b} \in [130, 140]$ GeV. We compare this region of statistical fluctuations in the asymmetry plot with a 1-dimensional slice of the correlation plot (lower right) for $p_{T,t} = 100 \pm 1$ GeV. The 1-dimensional distributions shows that in this range the normalized differential cross section has dropped below the visible range.

## 4 Outlook

We have shown that it is possible to GAN the full phase space structure of a realistic LHC process, namely top pair production all the way down to the kinematics of the six top decay jets. Trained on a simulated set of unweighted events this allows us to generate any number of new events representing the same phase space information. With the help of an additional MMD kernel we described on-shell resonances as well as tails of distributions. The only additional input was the final-state momenta related to on-shell resonances, and the rough phase space resolution of the on-shell pattern.

Our detailed comparison showed that relatively flat distributions can be reproduced at arbitrary precision, limited only by the statistics of the training sample. The mass values defining intermediate resonance poles were also easily extracted from the dynamic GAN setup. Learning the widths of the Breit-Wigner propagator requires an MMD kernel with sufficient resolution and is in our case only limited by the training time. The main limitation of the GAN approach is that statistical uncertainties in poorly populated tails of distributions in the training data appear as systematic uncertainties in the same phase space regions for the generated high-statistics samples. We have studied this effect in detail.

Because such a GAN does not require any event unweighting we expect it to be a useful

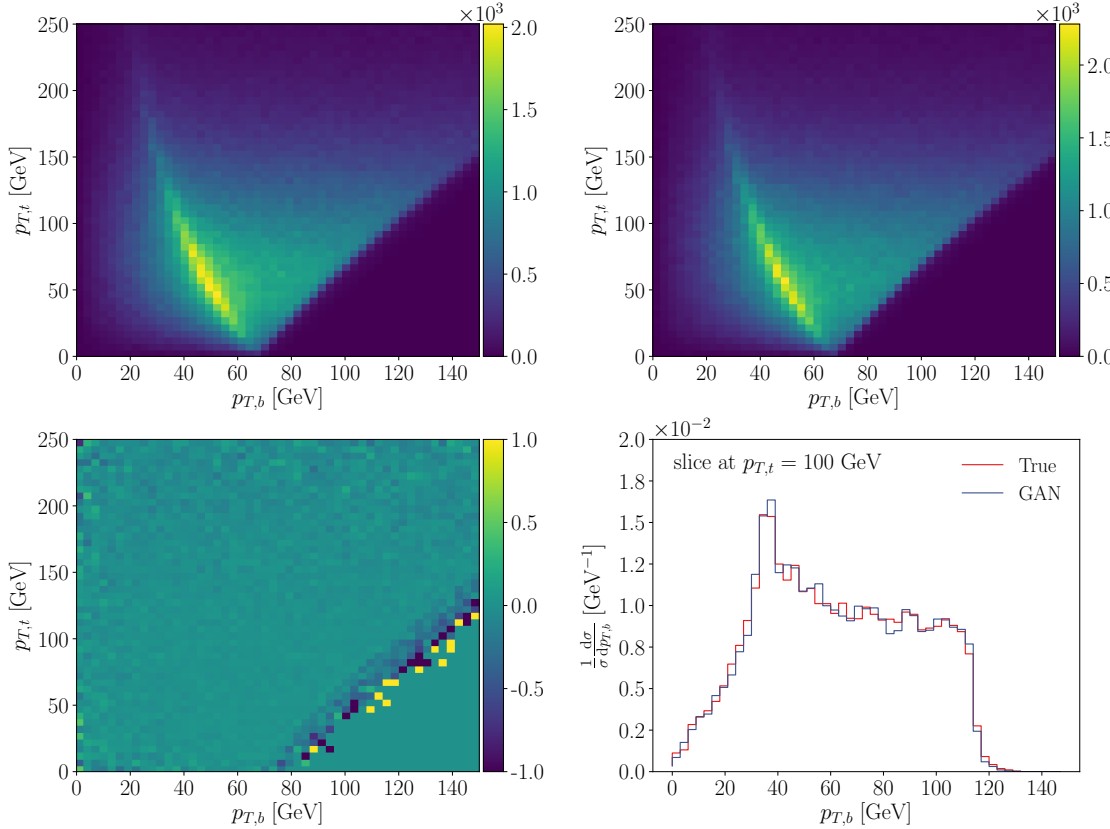

Figure 9: Correlation between $p_{T,t}$ and $p_{T,b}$ for the true data (upper left), GAN data (upper right) and the asymmetry between both (lower left). In addition, we show $p_{T,b}$ sliced at $p_{T,t} = 100 \pm 1$ GeV (lower right).

and fast[†] addition to the LHC event generation tool box. In case we want to improve the phase space coverage or include subtraction methods through a pre-defined event weight this is obviously possible. The same setup will also allow us to generate events from an actual LHC event sample or to combine actual data with Monte Carlo events for training, wherever such a thing might come in handy for an analysis or a fundamental physics question.

## Acknowledgments

We are very grateful to Gregor Kasieczka for his collaboration in the early phase of the project and to Jonas Glombitza and Till Bungert for fruitful discussions. We would also like to thank Steffen Schumann for very helpful physics discussions, asking all the right questions, and pointing out the similarity of on-shell peaks and phase space boundaries from a technical point of view. RW acknowledges support by the IMPRS-PTFS. The research of AB was supported by the Deutsche Forschungsgemeinschaft (DFG, German Research Foundation) under grant 396021762 — TRR 257 "Particle Physics Phenomenology after the Higgs Discovery".

---

[†]Once trained, our GAN generates 1 million events in 1.6 minutes on a laptop.

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
