# Peer review of "How to GAN LHC Events"

_SciPost Physics, doi:SciPost Phys. 7, 075 (2019)_

## Round 2 · Referee Report · Anonymous (Referee 1) · 2019-8-3

Strengths

1- Provides an interesting solution to generative modeling with localized features.

Weaknesses

1- Missing references to related papers. 2- Does not demonstrate that this method is learning genuinely new examples 3- Many of the claims in the conclusions are not substantiated with evidence in the body (e.g. "limited only by the statistics of the training sample" and "through a pre-defined event weight this is obviously possible")

Report

This study is a useful addition to the growing GANs for HEP literature in that it addresses the case of sharp features in phase space. There are a few points that I think need to be addressed before I could recommend the manuscript for publication - see the "Weaknesses" and "Requested changes".

Requested changes

1- 1903.02433 is a critical missing reference. Please also add a brief statement about how your paper is different than this one, which claims to do something similar to what you have done. 2- [2-5] Perhaps it would be nice to also cite ATL-SOFT-PUB-2018-001.
3- [6] Perhaps it would be nice to cite 1701.05927 (which uses a GAN), 1807.03685, and 1804.09720, which all are deep neural network approaches to generating a parton shower. 4- Can you please demonstrate that your GAN is really able to generate statistically independent examples? If you really claim that it gets the full distribution correct, please show that it can model the tails as well as the bulk. You could maybe do this with bootstrapping to show that the statistical power of a GAN dataset that is 10x bigger than the training one is really 10x the one of the original dataset. My guess is that this will be true for the bulk, but not for the tails (in which case, perhaps you could modify your claims a bit). 5- Can you please substantiate the claims you make in the conclusions? (see above). 6- I did not fully understand the purpose of the lower panels in the bottom plots of Fig. 4. If the training stat. uncertainty is 100% in the tail, how can the GAN be within 20% of the true answer?

  • validity: high
  • significance: good
  • originality: good
  • clarity: good
  • formatting: excellent
  • grammar: good

Author:  Ramon Winterhalder  on 2019-10-01  [id 611]

(in reply to Report 1 on 2019-08-03)
Category:
answer to question

  1. [1903.02433][1] is a critical missing reference. Please also add a brief statement about how your paper is different than this one, which claims to do something similar to what you have done.

-> The paper is now cited and we added a brief comment in the introduction. They do not include intermediate particles and do not encounter any sharp phase space features.

  1. Perhaps it would be nice to also cite [ATL-SOFT-PUB-2018-001][2] (2-5).

-> It is cited now.

  1. Perhaps it would be nice to cite [1701.05927][3] (which uses a GAN), [1807.03685][4], and [1804.09720][5], which all are deep neural network approaches to generating a parton shower (6).

-> They are also cited now.

  1. Can you please demonstrate that your GAN is really able to generate statistically independent examples? If you really claim that it gets the full distribution correct, please show that it can model the tails as well as the bulk. You could maybe do this with bootstrapping to show that the statistical power of a GAN dataset that is 10x bigger than the training one is really 10x the one of the original dataset. My guess is that this will be true for the bulk, but not for the tails (in which case, perhaps you could modify your claims a bit).

-> We already say that not all regions are perfectly learned. We see a systematics effect due to low statistics of the training/batch data, which is described in the text. Furthermore, we show a correlation plot which shows that the full phase-space is covered. We have also checked carefully and that there are indeed no holes.

  1. Can you please substantiate the claims you make in the conclusions? Many of the claims in the conclusions are not substantiated with evidence in the body (e.g. "limited only by the statistics of the training sample" and "through a pre-defined event weight this is obviously possible")

-> We added some text in the body to clarify the training time claim of the resonances. The claim with the event weight is already explained int the discussion of the flat distributions.

  1. I did not fully understand the purpose of the lower panels in the bottom plots of Fig. 4. If the training stat. uncertainty is 100% in the tail, how can the GAN be within 20% of the true answer?

-> We have modified the discussion of the different panels and hope that it is clear now.

---

## Round 2 · Referee Report · Anonymous (Referee 2) · 2019-8-16

Strengths

1- It is important for the field to study new approaches to extend the simulation of events using deep generative models 2- This paper presents an interesting approach to model LHC events with GANs with the novel inclusion of a density measure to compare distributions (maximum mean discrepancy, MMD) to the loss function 3- Section 2 is a very readable introduction into Monte Carlo and phase space generation

Weaknesses

1- At present it is not clear if the claim to model the “full phase space of top events” can be justified 2- The main shortcomings of the paper are a missing discussion of overtraining (reproducing events similar to the training data) and a possible holes in the generated parameter space.
3- It is not clear how the performance of this approach compares to other approaches to generate LHC events with deep generative models

Report

This paper presents an interesting approach to model LHC events with GANs with the novel inclusion of a density measure to compare distributions (maximum mean discrepancy, MMD) to the loss function. The authors claim that they show that it is possible „to GAN the full phase space structure of a realistic LHC process, namely top pair production all the way down to the kinematics of the six top decay jets. „ At present it is not clear if this claim (especially “full phase space”) can be justified. It is found that a traditional GAN is non-optimal to generate correct event distributions and a density measure to compare distributions (maximum mean discrepancy, MMD) is added to the loss function. A detailed discussion of the ability of GANs (without MMD) to model such processes in the full phase space is currently missing and should be added.
The main shortcomings of the paper are a missing discussion of overtraining and a discussion of possible holes in the generated parameter space. This is needed to be able to understand the applicability of GANs for MC integration and to be able to compare different approaches. In the GAN approach the batch size dependence might be critical and the full density might only be learned if the batch size is very large. A discussion of the batch size dependence of the approach is missing in the paper and should be added. Finally, a discussion (or comparison) to other attempts in the literature is incomplete and would be helpful to understand the advantages of the setup proposed in this paper. A measure should be used to be able to compare different approaches (currently proposed on arxiv) in terms of learning the correct density and in terms of avoiding holes in the sampled parameter space. The work should be published when also a better justification and discussion of the PROs and CONs of this approach (also compared to others) can be added to a revised version.

Requested changes

- Title: The novelty of the paper is the MMD + GAN approach, this should be represented in the title (there will be various GAN papers on LHC events). Also the MMD approach could also be added to other generative models.
- Abstract: “which simply clone events and avoid highly inefficient event un-weighting.“
GANs do more than simply clone the training data and if this would be the case GANs cannot be used to help MC generation (one could simply use multiple times the same event). Maybe the editors mean something else with this sentence?
- Introduction: It is not clear how this paper is related to other work, e.g. reference 9 and 10, especially regarding the sentence „However, up to now high-dimensional phase space coverage including realistic multi-particle matrix elements has not been in reach of a GAN setup.” It should be explained why this is not solved already e.g. in reference 10. This paper also studies the same process (toptop production and decay to 6 objects) as reference 9. The difference of this approach compared to Ref 9 should also be discussed, i.e. the implementation of MMD in the loss function of the GAN (GAN-MMD), whereas reference 9 proposes to use a so-called density buffer for VAEs for density estimation.
- „Including higher- order corrections is obviously possible and should lead to ever higher gains in computing time. „ This is trivial if the training data includes higher order corrections, but otherwise (e.g. in terms of extrapolation or correction to leading order events) not shown and a highly difficult task. This is not clear to the reader.
- “instead clone reconstructed LHC events and use them to enhance analyses or to study features of the hard process.„ It is misleading (or at least not clearly defined) to say that a GAN “clones” events. A clone is just an identical copy. An objective of a generative model is not only to learn the ability to generate “clones”, but also “new” events similar to the training data by interpolation, see also the later comments on overtraining and “holes” in the parameter space.
Section 2:
- “ …us with 18 degrees of freedom „ This means that only the 3 vector is learned by the generative model and not the particle type or the particle mass. It is not clear why /how the particle type and mass can be assumed to be known if this is not a parameter of the problem ? A sentence should be added to clarify.
- Section 2.2: “induces a random distribution PG(x)….“ In the following the words distribution, event, batch etc. are used without definition, e.g. it is not clear to a reader if “x” is a 15-dimensional set of numbers representing an “event” or if it a 15-dimensional set of random input into the generator network and then P(x) is the “output” (i.e. the event or a batch ?). The correspondence between “distribution” and “batch” and “the total sample of random inputs x” , the output of the GAN etc. needs to be clearly defined.
- Furthermore, it is said that the “the discriminator network compares two data sets, the true distribution PT (x) and the generated distribution PG(x). „ In the loss function eq. 10 the discriminator compares the two probability distributions event by event x or as a batch. The description in the section is not clear, especially since MMD is not introduced yet.
- In the following the „regularized Jensen-Shannon GAN“ is used as defined in Ref. 14. This should be stated.
- VAEs… -> “latent modelling and the marginalization of unnecessary variables“. Finding the best variables (also using latent space modelling) is a very relevant problem for LHC and VAEs may have advantages due to naturally avoiding e.g. mode collapse problems of GANs or in terms of overtraining.
- Section 2.3 : In contrast to the number of parameters stated at the beginning of section 2 now the mass of intermediate particles are given to the network in addition. Could this be avoided if the 4-vector would be given as input and why was this not done ? The input data should explicitly be defined, e.g. in a Table.
- The paper combines GAN and MMD. Similar has been done before e.g. in Generative moment matching networks (GMMN, arxiv:1502.02761 and others). Citations should be added and previous work on MMD + GAN combination should be discussed in the paper.
- The authors should state training time, number of training data etc. It would also be good to provide a link to the code on github to be able to compare the results with other approaches.
- Page 9: flat distributions: Ref [9] claims that flat distributions with sharp physical bounds (like the phi angle) are very difficult to reproduce with GANs. To have the ability to compare the different approaches proposed so far it is necessary to see the e.g. phi distributions of the 6 objects and e.g. correlations like phi_bjet vs phi_lepton for a GAN (without MMD term in loss) and the MMD improved GAN (which might be able to solve the problem) ?
- Non-flat distributions: It should be noted (Figure 5) that also with MMD the correct distribution is not learned perfectly (the true distribution is much narrower). There are still large differences and a ratio plot would help to quantify the differences.
- The networks have a large complexity and overtraining is not discussed in the paper, i.e. how much are the produced events copies of the training data. A distribution similar to ref. 9 might be helpful (phi_1 vs phi_2) to see possible holes in the generated phase space. Figure 7 is interesting in this respect and here a slice with 100 +-10 GeV is plotted. To see holes in the phase space it would be interesting to see the distribution for much smaller slices (e.g. +- 1 or +- 0.01 GeV). This plot should be compared to the training data.

Details:
Page 7: formulation : „If the resonance and with it the kernel width „

  • validity: good
  • significance: high
  • originality: good
  • clarity: good
  • formatting: good
  • grammar: good

Author:  Ramon Winterhalder  on 2019-10-01  [id 612]

(in reply to Report 2 on 2019-08-16)

  1. Title: The novelty of the paper is the MMD + GAN approach, this should be represented in the title (there will be various GAN papers on LHC events). Also the MMD approach could also be added to other generative models.

-> We slightly disagree with that judgement - the MMD is a technical aspect which allows us to describe realistic phase space configurations with a GAN. In that sense we think the title should be fine.

  1. Abstract: “which simply clone events and avoid highly inefficient event un-weighting.“ GANs do more than simply clone the training data and if this would be the case GANs cannot be used to help MC generation (one could simply use multiple times the same event). Maybe the editors mean something else with this sentence?

-> We consistently changed clone to generate to make things more clear.

  1. Introduction: It is not clear how this paper is related to other work, e.g. reference 9 and 10, especially regarding the sentence „However, up to now high-dimensional phase space coverage including realistic multi-particle matrix elements has not been in reach of a GAN setup.” It should be explained why this is not solved already e.g. in reference 10. This paper also studies the same process (toptop production and decay to 6 objects) as reference 9. The difference of this approach compared to Ref 9 should also be discussed, i.e. the implementation of MMD in the loss function of the GAN (GAN-MMD), whereas reference 9 proposes to use a so-called density buffer for VAEs for density estimation.

-> We have added some detailed discussions of ither approaches and their benefits.

  1. „Including higher- order corrections is obviously possible and should lead to ever higher gains in computing time. „ This is trivial if the training data includes higher order corrections, but otherwise (e.g. in terms of extrapolation or correction to leading order events) not shown and a highly difficult task. This is not clear to the reader.

-> We added a sentence to make clear, that we consider the trivial case in which the higher-orders are included in the training data.

  1. “instead clone reconstructed LHC events and use them to enhance analyses or to study features of the hard process.„ It is misleading (or at least not clearly defined) to say that a GAN “clones” events. A clone is just an identical copy. An objective of a generative model is not only to learn the ability to generate “clones”, but also “new” events similar to the training data by interpolation, see also the later comments on overtraining and “holes” in the parameter space.

-> As before we replaced "clone" with "generate".

  1. Section 2: “ …us with 18 degrees of freedom „ This means that only the 3 vector is learned by the generative model and not the particle type or the particle mass. It is not clear why/how the particle type and mass can be assumed to be known if this is not a parameter of the problem? A sentence should be added to clarify.

-> We assume that the type of external particles and hence their masses are known once we specify a physics process. We are not quite sure what the referee means here.

  1. Section 2.2: “induces a random distribution PG(x)….“ In the following the words distribution, event, batch etc. are used without definition, e.g. it is not clear to a reader if “x” is a 15-dimensional set of numbers representing an “event” or if it a 15-dimensional set of random input into the generator network and then P(x) is the “output” (i.e. the event or a batch ?). The correspondence between “distribution” and “batch” and “the total sample of random inputs x” , the output of the GAN etc. needs to be clearly defined.

-> We clarified meaning and correlations of the different terms in the body. It should be clear now what is an input and output. We also describe this in Fig. 3.

  1. Furthermore, it is said that the “the discriminator network compares two data sets, the true distribution PT (x) and the generated distribution PG(x). „ In the loss function eq. 10 the discriminator compares the two probability distributions event by event x or as a batch. The description in the section is not clear, especially since MMD is not introduced yet.

-> We slightly changed the wording now. However, it is batchwise and follows Eqs.(10) and (11).

  1. In the following the „regularized Jensen-Shannon GAN“ is used as defined in Ref. 14. This should be stated.

-> We now mention and cite the „regularized Jensen-Shannon GAN“.

  1. VAEs… -> “latent modelling and the marginalization of unnecessary variables“. Finding the best variables (also using latent space modelling) is a very relevant problem for LHC and VAEs may have advantages due to naturally avoiding e.g. mode collapse problems of GANs or in terms of overtraining.

-> We do not use VAEs, so the reader should rely on the excellent descriptions for instance in our Ref.[13].

  1. Section 2.3 : In contrast to the number of parameters stated at the beginning of section 2 now the mass of intermediate particles are given to the network in addition. Could this be avoided if the 4-vector would be given as input and why was this not done ? The input data should explicitly be defined, e.g. in a Table.

-> The mass of the intermediate particles is not explicitly given to our network, as described in Sec. 2.3.

  1. The paper combines GAN and MMD. Similar has been done before e.g. in Generative moment matching networks (GMMN, arxiv:1502.02761 and others). Citations should be added and previous work on MMD + GAN combination should be discussed in the paper.

-> Citations and a short comment have been added.

  1. The authors should state training time, number of training data etc. It would also be good to provide a link to the code on github to be able to compare the results with other approaches.

-> We do not offer a github link right now, because our code is not ready to be published. All other information are now included in Tab. 1.

  1. Page 9: flat distributions: Ref [9] claims that flat distributions with sharp physical bounds (like the phi angle) are very difficult to reproduce with GANs. To have the ability to compare the different approaches proposed so far it is necessary to see the e.g. phi distributions of the 6 objects and e.g. correlations like phi_bjet vs phi_lepton for a GAN (without MMD term in loss) and the MMD improved GAN (which might be able to solve the problem) ?

-> We already show a correlation plot including sharp boundaries in Fig. 7.

  1. Non-flat distributions: It should be noted (Figure 5) that also with MMD the correct distribution is not learned perfectly (the true distribution is much narrower). There are still large differences and a ratio plot would help to quantify the differences.

-> The difference is already obvious in the plot and the representation has been chosen to see the improvement from the MMD. We also state that even with the MMD the match is not yet perfect and give possible ways to improve this.

  1. The networks have a large complexity and overtraining is not discussed in the paper, i.e. how much are the produced events copies of the training data. A distribution similar to ref. 9 might be helpful (phi_1 vs phi_2) to see possible holes in the generated phase space. Figure 7 is interesting in this respect and here a slice with 100 +-10 GeV is plotted. To see holes in the phase space it would be interesting to see the distribution for much smaller slices (e.g. +- 1 or +- 0.01 GeV). This plot should be compared to the training data.

-> We have modified the correlation plot such that we have smaller slices of +- 1 GeV now. It can be seen that there are no holes in this representation. The same is true for other distributions and correlations, even though we only show the correlation we find most interesting.

  1. Details: Page 7: formulation : „If the resonance and with it the kernel width „

-> We changed this.

---

## Round 3 · Referee Report · Anonymous · 2019-10-3

Report

This is a followup report, now considering v2. Thank you to the authors for addressing my comments on v1. I now only have two followup points:

- Fig. 4: I still don't understand how the GAN can do better (closer to the true distribution) than the stat. uncertainty on the training dataset. Please explain.

- v1 comment: Can you please demonstrate that your GAN is really able to generate statistically independent examples? If you really claim that it gets the full distribution correct, please show that it can model the tails as well as the bulk. You could maybe do this with bootstrapping to show that the statistical power of a GAN dataset that is 10x bigger than the training one is really 10x the one of the original dataset. My guess is that this will be true for the bulk, but not for the tails (in which case, perhaps you could modify your claims a bit).

Your answer: We already say that not all regions are perfectly learned. We see a systematics effect due to low statistics of the training/batch data, which is described in the text. Furthermore, we show a correlation plot which shows that the full phase-space is covered. We have also checked carefully and that there are indeed no holes.

Followup: Perhaps I should say this another way: you are advocating that people can use your tool to augment physics-based simulations. If I have a simulator, I could use your method to make e.g. 10x the number of events I started with. In order for me to believe that this is a useful exercise, you need to convince me that the 10x more events I got are statistically independent from the original physics-based simulation. If they are not, then I have not gained with the GAN. In my first comment, I proposed a way to show this, but there may be other ways to convince the reader.

  • validity: -
  • significance: -
  • originality: -
  • clarity: -
  • formatting: -
  • grammar: -

Author Ramon Winterhalder on 2019-11-07
(in reply to Report 1 on 2019-10-03)
Category:
answer to question

Fig. 4: I still don't understand how the GAN can do better (closer to the true distribution) than the stat. uncertainty on the training dataset. Please explain.

We don't say that the GAN does better than the stat. uncertainty. If the stat. Uncertainty is 20% the GAN obviously can only be equally precise. However, stating the GAN is correct within 10% means, that the ratio of GAN/True is 0.9. Considering the stat. uncertanty of the trainings data the GAN agrees with the true events within this uncertainty!

v1 comment: Can you please demonstrate that your GAN is really able to generate statistically independent examples? If you really claim that it gets the full distribution correct, please show that it can model the tails as well as the bulk. You could maybe do this with bootstrapping to show that the statistical power of a GAN dataset that is 10x bigger than the training one is really 10x the one of the original dataset. My guess is that this will be true for the bulk, but not for the tails (in which case, perhaps you could modify your claims a bit). Your answer: We already say that not all regions are perfectly learned. We see a systematics effect due to low statistics of the training/batch data, which is described in the text. Furthermore, we show a correlation plot which shows that the full phase-space is covered. We have also checked carefully and that there are indeed no holes. Followup: Perhaps I should say this another way: you are advocating that people can use your tool to augment physics-based simulations. If I have a simulator, I could use your method to make e.g. 10x the number of events I started with. In order for me to believe that this is a useful exercise, you need to convince me that the 10x more events I got are statistically independent from the original physics-based simulation. If they are not, then I have not gained with the GAN. In my first comment, I proposed a way to show this, but there may be other ways to convince the reader.

We now show a 2D correlation plot of phi_j1 vs phi_j2 for 1 million true events and 1/10/50 million generated events next to it with a very small binning. This shows, that the GAN truly populates all phase space regions beyond the training data and does not produce any holes. Further, statistical independence is also a priori enforced by sampling from random noise.

---

## Round 3 · Referee Report · Anonymous · 2019-10-28

Strengths

see previous report

Weaknesses

see previous report and detailed requests below

Report

see previous report and detailed requests below

Requested changes

Comments to arxiv version 3

Main concerns:

- The authors claim that they were able to sample the "full phase space" of the ttbar process. There is no indication that the GAN can sample the phase space of the samples without "holes". In addition, there is no discussion of how much of the phase space the GAN can sample outside the training data. The training data is a rather small subset of the true high dimensional phase space. This could be visualized by producing many more events through the GAN than were used as training data, and by displaying the sampled phase space with very small bin resolutions, revealing the granularity of the training data. It cannot therefore be concluded that the GAN scans the "full phase space". The training data and the capacity of the GAN are huge. It is not clear what we learn beyond the 1 Million training data events.
- Showing e.g. phi_object1 vs phi_object2 with very small bin sizes could be a way to show how the GAN is able to fill the “holes” in the high-dim phase space beyond the training data. It would also be a way to see how much mode-collapse is a really avoided.
- I like to repeat the demand to show the phi distributions of all 6 objects. It is interesting to show that they are indeed flat given the claim by reference 14.
- It is not clear how essential the MMD term is to reproduce the distributions. Also, the effect of the MMD term on the phi, eta and pt distributions should be shown.
- The details of the MMD configurations should be added to the draft, i.e. which kernels, widths etc. have been used ?
- Since the authors do not want to state the use of MMD in the title, I would recommend to mention this at least in the abstract.
- Code should be released with this publication. At least, the data produced by the GAN and the training data should be made available. The results are otherwise hardly reproducible.

Details:
- „page 5 „for each point“  I ssume you mean for each „batch“

  • validity: -
  • significance: -
  • originality: -
  • clarity: -
  • formatting: -
  • grammar: -

Author Ramon Winterhalder on 2019-11-07
(in reply to Report 2 on 2019-10-28)
Category:
answer to question
reply to objection

The authors claim that they were able to sample the "full phase space" of the ttbar process. There is no indication that the GAN can sample the phase space of the samples without "holes". In addition, there is no discussion of how much of the phase space the GAN can sample outside the training data. The training data is a rather small subset of the true high dimensional phase space. This could be visualized by producing many more events through the GAN than were used as training data, and by displaying the sampled phase space with very small bin resolutions, revealing the granularity of the training data. It cannot therefore be concluded that the GAN scans the "full phase space". The training data and the capacity of the GAN are huge. It is not clear what we learn beyond the 1 Million training data events.

We show a correlation plot which is most interesting structure wise, which does not show any holes in the phase-space. Further, upon request, we now added a 2D correlations of phi_j1 vs phi_j2 for 1 million true events and 1/10/50 million generated events next to it with a very small binning.

Showing e.g. phi_object1 vs phi_object2 with very small bin sizes could be a way to show how the GAN is able to fill the “holes” in the high-dim phase space beyond the training data. It would also be a way to see how much mode-collapse is a really avoided.

We now show those plots and dont see any "holes" in the phase space.

I like to repeat the demand to show the phi distributions of all 6 objects. It is interesting to show that they are indeed flat given the claim by reference 14.

We slightly disagree on the importance of those distributions as they do not show any interesting physics and are indeed flat. However, we now show the phi distributions of for two arbitrary objects which are indeed flat as expected. We restricted ourselves to show only two of them as they look all the same and do not encode any other interesting information.

It is not clear how essential the MMD term is to reproduce the distributions. Also, the effect of the MMD term on the phi, eta and pt distributions should be shown.

We don't see any effect of the MMD on any other observable than the invariant masses, hence we only show the effect of the MMD on the invariant mass distributions. For the invariant masses, we show 4 plots which clearly indicate the importance of the MMD in resolving the sharp local features.

The details of the MMD configurations should be added to the draft, i.e. which kernels, widths etc. have been used ?

The details of the MMD are given both, in the corresponding plots and the describing text along with the plots.

Since the authors do not want to state the use of MMD in the title, I would recommend to mention this at least in the abstract.

We now mention the MMD in the abstract.

Code should be released with this publication. At least, the data produced by the GAN and the training data should be made available. The results are otherwise hardly reproducible.

Upon request we will happily share our code and training data. However, we decided against making it publicly available on GitHub.

Details: „page 5 „for each point“ . I ssume you mean for each „batch“

No, the complete phrase is "for each point in a batch..", hence, the wording is correct.

---

## Round 4 · Referee Report · Anonymous (Referee 1) · 2019-11-10

Report

Thank you for your response! I would like to quickly followup on both points - hopefully it won't take much time to do one last iteration before acceptance.

You wrote

My question: Fig. 4: I still don't understand how the GAN can do better (closer to the true distribution) than the stat. uncertainty on the training dataset. Please explain.

Your answer: We don't say that the GAN does better than the stat. uncertainty. If the stat. Uncertainty is 20% the GAN obviously can only be equally precise. However, stating the GAN is correct within 10% means, that the ratio of GAN/True is 0.9. Considering the stat. uncertanty of the trainings data the GAN agrees with the true events within this uncertainty!

My response: This doesn't make sense to me - if your stat. uncertainty is 20%, then on average, the GAN cannot agree with the true to within 10%. The fluctuations in GAN/True better be comparable or larger than the stat uncertainty. Either I am completely missing the point or what is stated is not correct. In either case, please clarify in the text!

v1 comment: Can you please demonstrate that your GAN is really able to generate statistically independent examples...

Your answer: We now show a 2D correlation plot of phi_j1 vs phi_j2 for 1 million true events and 1/10/50 million generated events next to it with a very small binning. This shows, that the GAN truly populates all phase space regions beyond the training data and does not produce any holes. Further, statistical independence is also a priori enforced by sampling from random noise.

My response: Thank you for this test! However, I don't think this exactly answers my question. First of all, your samples need not be statistically independent from the training sample because you are starting from random noise. The GAN could literally be memorizing and picking randomly one of the events from the training set. I think what you want to do is to also have a sample with 50M true events and show that the GAN with 50M looks like that. It is nice to see that there are no obvious holes, though there is clearly some feature in the bottom right plot of Fig. 6 between the yellow bands that looks to not be in the original. I won't insist too much on this point, so maybe you can at least partially convince the reader with some slight modifications to the text.

---

## Round 4 · Referee Report · Anonymous (Referee 3) · 2019-11-11

Strengths

1- Clarity of writing 2- Novelty of approach

Weaknesses

1- The case for the importance of this work should be strengthened

Report

The paper presents a study of using GANs to generate simulated particle collisions, including heavy particle production and multi-step decay. They are able to map the high-dim phase space with impressive accuracy (for a GAN), which to my knowledge has not been done before, especially in the context of tricky kinematics effects such as narrow resonances and phase-space boundaries.

The paper is also very well written, with clear, crisp prose and a good balance of references and introductory text to guide the reader.

There are two important questions in my mind: is the paper correct, and is it important? I address those below, and then comment on the discussion of the other referees.

(1) Is the work correct?

The paper is technically impressive, and to my reading the central claims made are well supported.

I have one worry. It would be nice to ensure the generator isn't memorizing the dataset, because this would just be unweighting a bunch of weighted events with a lot of extra steps involved. In image analyses, one often sees pictures of generated samples juxtaposed with the nearest neighbor in the true data to make sure that they are sufficiently different. (See the discussion in https://arxiv.org/pdf/1706.08224.pdf about support size of a generator; Figure 1 is the type of figures we have in mind). Alternatively, one could just measure the expected distance to the nearest point in the true dataset over the true dataset, and then calculate the expected distance to the nearest neighbor in the true dataset over the generated dataset to see that they are comparable. If generated samples are typically much closer to something in the true dataset, then it is clearly just memorizing data points.

(2) Is the work important/relevant?

Not every correctly-done study is worthy of publication. It needs to add something new and relevant.

What they have done is new IMO, but the case for the importance and relevance of the work presented here needs clarification.

The authors correctly point out that simulation tools are essential for likelihood-free inference in HEP, because we do not have good theoretical control of showers/hadronization/detector response and are forced to use simulations. They are also correct that the the detector response is by far the slowest and least-well controlled element of this chain, and that alternative approaches like GANs offer valuable speed-ups. They give a fair summary of recent progress in these areas.

But this paper does not focus on those areas that critically need to be sped up, and where theoretical knowledge is limited. Instead, this paper focuses on the simulation of the hard scattering, the piece that is both very well theoretically controlled and already extremely fast. They do not argue why such a tool is useful or important.

Why would HEP want to replace a procedure which is deeply connected to the underlying theory (drawing events from the PDF calculated by the matrix element) and can be automatically generated for an arbitrary new theory with a more-black-box GAN? They claim their GAN is fast (footnote on pg 15), but do not provide a straightforward comparison with existing tools, so this seems unlikely to be their primary argument. They emphasize that their GAN produces unweighted events rather than weighted events, but this hardly seems enough to motivate this complex procedure. They suggest that their GAN could also be generalized to learn the more useful steps of showers/hadronization/detector response, but do not show any results there, and seem to dismiss it as less interesting?

Perhaps there is an argument to be made for the importance and relevance of this work, but to my reading the authors do not make it. My best guess is that the authors found this task to be unsolved and a fun intellectual challenge.

I don’t think this should prevent the publication of this work, but my advice to the authors is to consider adding a paragraph to the introducing making a stronger case for the relevance of this. To wit: why should someone use this tool rather than use MG5?

(3) Have the authors addressed technical concerns raised by the other reviewers?

  • I find the author’s replies to be satisfactory. On the most significant questions, I think they have clarified their claims of statistical uncertainty (as being relative to the generator sample, not the truth) and demonstrated that they do not have holes in their generated phase space.

Requested changes

1- I recommend (but do not require) adding discussion to the introducing making a stronger case for the relevance/importance of this.
2- I would like to see more evidence that the generator isn't memorizing the dataset.

---

## Round 4 · Referee Report · Anonymous (Referee 2) · 2019-11-29

Report

see requested changes below.

Requested changes

Dear all,

I do not know why the refereeing procedure for this paper was closed and this paper has been accepted already.

I like to add (I was a referee for the first and second round of comments) that I strongly recommend that all code needs to be made public.
This paper has a very strong claim:
The claim is that this GAN is an excellent model for up to 20-dimensional density estimations (di-top events) since this paper describes that all distributions and correlations (beside the peaks where MMD was used in 1d) can be described by the GAN model.

To my knowledge, in the ML community GANs are till today not regarded as good models for density estimation. If GANs can make almost perfect models for density estimation, then this is a very important result.

Therefore, the community needs to be able to reproduce this result. The generator NN including the determined weights, the training scripts, training data and the model data should be made public, e.g. via github.
If it works, this GAN is very much in demand in the community.

---

## Editorial Decision

published